# Towards Strengthening the Resilience of IoV Networks—A Trust Management Perspective

**Yingxun Wang** [1,2,*], **Hushairi Zen** [1], **Mohamad Faizrizwan Mohd Sabri** [1], **Xiang Wang** [1,2] **and Lee Chin Kho** [1]

1 Faculty of Engineering, Universiti Malaysia Sarawak, Kota Samarahan 94300, Sarawak, Malaysia; zhushair@unimas.my (H.Z.); msmfaizrizwan@unimas.my (M.F.M.S.); xwang@qlit.edu.cn (X.W.); lckho@unimas.my (L.C.K.)
2 Faculty of Mechanical and Electrical Engineering, Qilu Institute of Technology, Jinan 250200, China
* Correspondence: wyx8586@qlit.edu.cn

**Abstract:** Over the past decade or so, considerable and rapid advancements in the state of the art within the promising paradigms of the Internet of Things (IoT) and Artificial Intelligence (AI) have accelerated the development of conventional Vehicular Ad Hoc Networks (VANETS) into the Internet of Vehicles (IoV), thereby bringing both connected and autonomous driving much closer to realization. IoV is a new concept in the Intelligent Traffic System (ITS) and an extended application of IoV in intelligent transportation. It enhances the existing capabilities of mobile ad hoc networks by integrating them with IoT so as to build an integrated and unified vehicle-to-vehicle network. It is worth mentioning that academic and industrial researchers are paying increasing attention to the concept of trust. Reliable trust models and accurate trust assessments are anticipated to improve the security of the IoV. This paper, therefore, focuses on the existing trustworthiness management models along with their corresponding trust parameters, as well as the corresponding trust evaluation parameters and simulation, which provide the basis for intelligent and efficient model suggestions and optimal parameter integration. In addition, this paper also puts forward some open research directions that need to be seriously solved before trust can play its due role in enhancing IoV network elasticity.

**Keywords:** vehicular ad hoc networks; internet of vehicles; intelligent traffic system; trustworthiness management models; trust evaluation; smart cities

## 1. Introduction

With the rapid advancement in the automotive industry, the number of vehicles in countries around the world is increasing at an unprecedented rate, resulting in severe challenges to the existing transportation system. On the one hand, traffic accidents occur frequently, hence, putting the lives of vehicular passengers and pedestrians at risk. The World Health Organization (WHO) released a report on road safety, stating that every year, around 1.35 million people die in road traffic accidents, and an average of one person dies in a traffic accident every 24 s. On the other hand, the issue of traffic congestion is becoming serious, not only restricting the orderly progress of urban traffic but also causing huge economic losses and environmental pollution. The Intelligent Transportation System (ITS) came into existence in order to reduce traffic accidents and alleviate traffic congestion. As an important component of smart cities, it paves the way for future transportation systems. The ITS combines data transmission technology, electronic sensing technology, information technology, computer technology, and various control technologies to build an efficient and real-time integrated traffic management system.

In recent years, wireless networking technology has matured. In particular, with the rapid development of long-term evolution (LTE) and fifth-generation (5G) mobile communication technology, the development of the concept of Internet of Vehicles (IoV) has attracted the attention of researchers. The use of 5G can facilitate vehicular networks

in having a more stable and low latency communication, and accordingly, the notion of 5G-ITS has appeared in the literature. In [1], Carlos proposed a 5G V2X (vehicle to everything) ecosystem for the IoV. The proposed ecosystem was based on the notion of a Software-Defined Network (SDN), and it mainly used 5G to evaluate the video Internet and vehicle-to-vehicle communication in rural and urban vehicle environments. In [2], Benjamin described the security and privacy challenges of using 5G technology in IoV and analyzed solutions to different types of attacks from a technical level. In [3], a crowdsensing system based on blockchain was proposed by Wang to ensure the security and privacy of 5G IoV. A deep reinforcement learning algorithm was adopted, and the effectiveness and security of the system were verified by experiments in a computer with a 64-bit Windows 10 operating system.

As an important part of the ITS, the Vehicular Ad Hoc Network (VANET) integrates the Mobile Internet and the Internet of Things (IoT), as described in Figure 1. As a special Mobile Ad Hoc Network (MANET), it has the characteristics of openness, high mobility and dynamic change in the topology structure of the network.

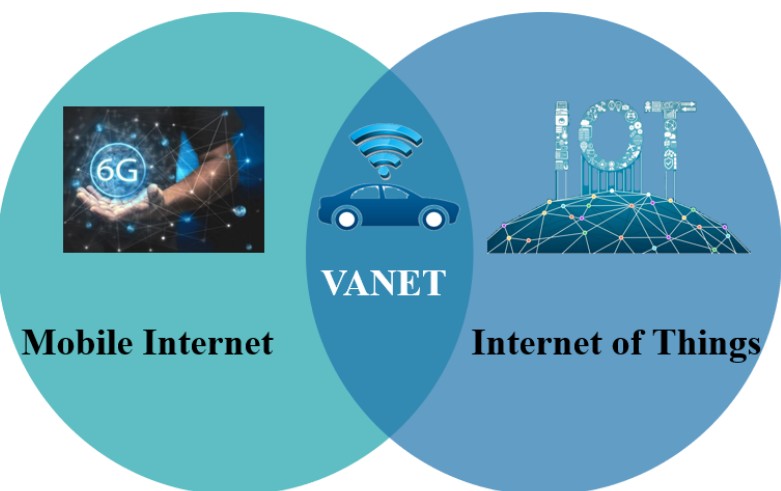

**Figure 1.** The relationship between the VANET and the Internet.

VANETs sense the surrounding road environment through on-board sensors. With the development of information and communication technology, VANETs realize network connections between vehicles, vehicles and the roadside, vehicles and pedestrians, and vehicles and service platforms; support traffic information sharing and vehicle–vehicle cooperation; and greatly improve road traffic safety and efficiency. A simple VANET is composed of an on-board unit (OBU), a roadside unit (RSU), and a trusted authority (TA). OBU refers to the communication device integrated in the vehicle. When a vehicle passes through RSU at high speed, OBU and RSU communicate via one of the radio access technologies. According to the technical reports of the 3rd Generation Partnership Project (3GPP) [4], the main business categories of IoVs are:

- *Driving Safety*—Driving safety is the most important task of the IoV and is of great significance for protecting people. It mainly includes collision warning, emergency vehicle warning, dangerous road conditions warning, and automatic driving;
- *Traffic Efficiency*—Traffic efficiency is mainly aimed at alleviating urban traffic congestion and providing green, efficient, and comfortable travel services for people. It mainly includes traffic light control, adaptive cruise, and vehicle choreography;
- *Information Services*—Information services mainly provide infotainment services, high-precision map download, navigation, and other value-added services. These include multimedia entertainment, high-precision map download, and remote vehicle diagnosis. Therefore, VANETs have huge social and commercial value.

In VANETs, the vehicles mainly act as nodes, propagating information; however, VANETs suffer from wireless instability, cover a relatively small mobile network, and are only suitable for local and discrete environments. On the contrary, IoV can process the messages it receives based on its perception ability. Every vehicle is regarded as an intelligent object in IoV and is equipped with a powerful multi-sensor platform, communication technology, computing units, and IP-based direct or indirect connections, thereby enabling it to connect to the Internet and other networking entities. In this regard, cellular vehicle to everything (C-V2X) is a promising vehicular communication technology that employs a cellular network, as shown in Figure 2.

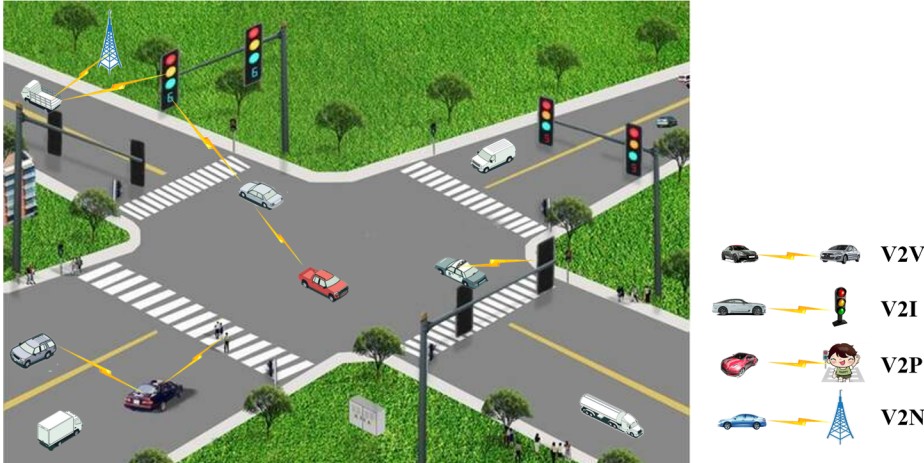

**Figure 2.** C-V2X schematic.

However, several security issues hinder the realization of IoV. Traditional security technologies based on authentication and encryption mechanisms cannot effectively identify internal attacks initiated by authenticated nodes. The core idea of a trust management mechanism is to ascertain the trust value of nodes and then identify malicious nodes by making intelligent decisions. The concept of 'trust management' was first proposed by Blaze [5] by providing a precise definition of trust and predicting the expected trust based on the probability distribution method. However, this model cannot effectively handle malicious attacks on the trust management mechanism itself, because it uses a simple weighted average method when calculating trust. In [6], the author put forward the concept of "scope" in trust management and can calculate the value of "scope", and they define another method to calculate the trust value. Many scholars have since begun to establish various trust models with different trust evaluation algorithms for different network environments, such as the P2P network, social networks and ad hoc networks. Classical trust models include Eigen Trust [7], Power Trust [8], Peer Trust [9], and Trust Guard [10]. Recently, many experts and scholars have carried out active research on the trust evaluation technology of IoV, and they obtained some interesting results.

Tan [11] put forward a trust management mechanism to ensure data security, which can help to choose the best path for a route. Li [12] proposed an attack-resistant trust management scheme for the IoV, which can resist false feedback attacks and switch attacks. Hu [13] put forward a trust evaluation scheme, REPLACE, at the scene of highway vehicle arrangement, which was used to help the user vehicles in the motorcade choose the trusted head vehicle to follow. Soley [14] proposed a trust model for IoV based on fuzzy logic and fog nodes, which made use of the characteristics of fuzzy logic. This model was good at handling inaccurate messages and had strong fault tolerance to evaluate the trust of messages transmitted in IoV. Lin [15] introduced social relations into the Internet of Vehicles and put forward the concept of "Vehicles Social Networks" (VSNs), which reflected the influence of social characteristics and human behavior on VANET. Chaker [16] proposed introducing vehicle roles to help allocate initial trust values. Xiao [17] introduced the

PagePank algorithm, which was originally used for web page sorting. According to the local trust, a relatively stable local trust link graph that was independent of the dynamic network topology structure was constructed, and then, the global trust of nodes was iteratively calculated based on the local trust link graph. In [18], an evaluation method of trust based on blockchain technology was proposed, which could resist false message attacks.

### 1.1. Contributions of the Paper

The main contributions are summarized as follows:

- To clarify the development process from VANET to IoV firstly. Owing to certain limitations of VANET, IoV has developed rapidly due to its special advantages.
- This paper conducts comprehensive research on the trust management in the Internet of Vehicles, including the trust model, trust parameters, simulation methods and model evaluation methods. In addition, various security attacks are classified and explained.
- Finally, the future research direction of the IoV trust management mechanism is put forward, which lays the foundation for follow-up research.

### 1.2. Outline of the Paper

The remainder of this paper is organized as follows. In Section 2, the development history and advantages of VANET are described in detail, and more importantly, the limitations of VANET at present are put forward. The appearance of IoV solves the limitations of VANET, and the architecture and key technologies of IoV are introduced in detail. Section 3 mainly introduces the trust management in IoV. By consulting the literature, we summarize all the trust classifications, trust models, trust parameters, evaluation parameters, and the classification of security attacks. In Section 4, future research directions for IoV and the problems to be solved are introduced.

## 2. From Vehicular Ad Hoc Networks to Internet of Vehicles

### 2.1. Vehicle Ad Hoc Networks (VANETS)

The MANET is one of the important components of the Wireless Ad Hoc Network (WANET). From the perspective of computer technology, MANET is defined as the working group of mobile ad hoc networks. There are two types of ad hoc network: fixed nodes and mobile nodes. MANET refers to an ad hoc network with mobile nodes, and it comprises:

- *Vehicle Ad Hoc Networks (VANETs)*—VANETs realize the coordination of vehicles, people, and roads through V2X, which is an artificial intelligence technology that can help vehicles take actions in an intelligent way when they encounter danger. Its appearance has greatly improved traffic safety and traffic efficiency;
- *Smart Phone Ad Hoc Networks (SPANs)*—SPANs create peer-to-peer networks using existing hardware devices, independent of cellular operator networks, wireless access points or traditional network infrastructure;
- *Internet-based Mobile Ad Hoc Networks (IMANETs)*—IMANETs support internet protocols such as TCP/UDP and IP;
- *Hub-Spoke MANET*—A plurality of sub-mobile ad hoc networks can be connected to the traditional central radiating VPN to create geographically distributed mobile ad hoc networks;
- *Military or Tactical Wireless Ad Hoc (MWANET)*—MWANET is a special self-organizing network for military departments, with special emphasis on data security, real-time requirements, data rate, radio range and integration, and fast routes in terms of mobility requirements;
- *Flying Ad Hoc Networks (FANETs)*—FANETs consist of drones, which can achieve great mobility and provide connectivity with remote areas.

VANET is a special commercial application of the traditional mobile ad hoc network. The evolutionary history of VANET is shown in Figure 3. VANET may be deployed by communication operators, content service providers, and government agencies, or jointly deployed by them, representing what is essentially a heterogeneous wireless network. C2CCC, the European vehicle-mounted communication alliance, provides a guiding architecture consisting of three domains, as shown in Table 1.

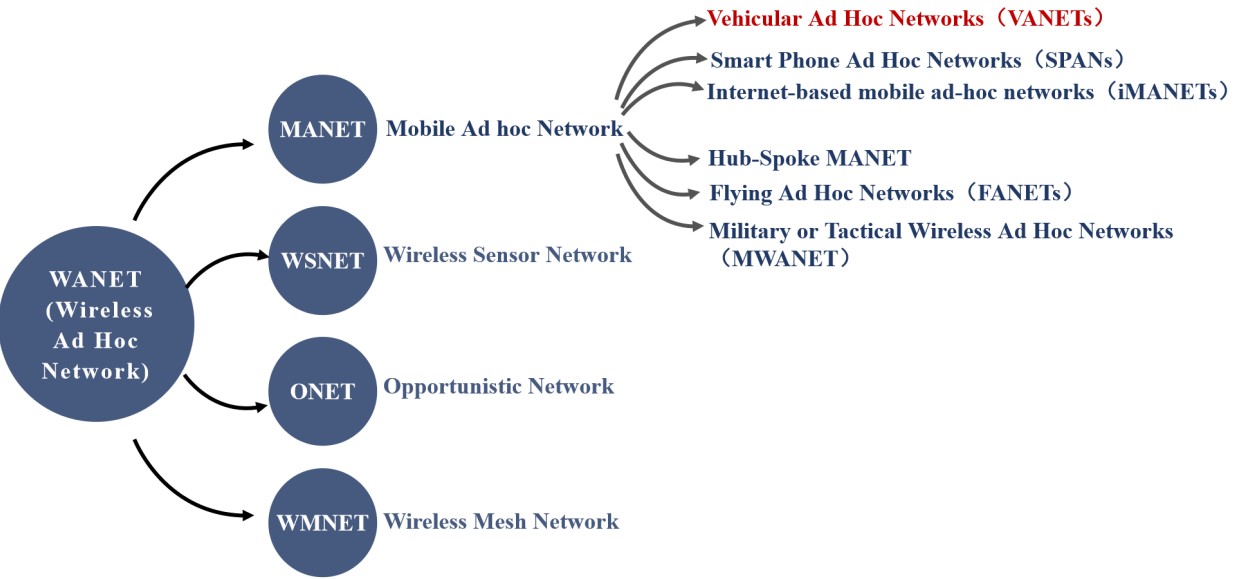

**Figure 3.** Evolution history of the VANET.

**Table 1.** Three domains of VANET.

| Domain | Function |
|---|---|
| In-Vehicle Domain | Communication between the OBU and application units (AUs) within the vehicle. AU can be a specific device, such as a mobile phone, or a virtual module integrated into an OBU. The connection can be wired or wireless, such as WiFi or Bluetooth. |
| Ad Hoc Domain | The wireless communication between OBUs and RSUs can be single-hop or multi-hop, namely V2V and V2R. |
| Infrastructure Domain | OBU and RSU communicate with infrastructure components, such as satellite, hot spot, and 5G, to access the Internet. For RSUs, the connection can be wired. |

VANET has some advantages because its network nodes are all vehicles and roadside devices:

- The vehicle can achieve sufficient energy support, and the carrying space of the vehicle can also ensure the good performance of its wireless communication equipment. At the same time, its storage and computing capabilities are also very powerful. In the same way, the traffic facilities on both sides of the road also have sufficient energy supply, and the computing storage capacity and wireless communication capacity are guaranteed;
- The popularity of GPS and GIS (Geographic Information System) makes VANET rich in external auxiliary information, including not only its own location information but also the geographical information of its area, such as road direction, traffic light distribution, etc.;

- As compared with other MANETs, the nodes of VANET move more regularly. The network topology is relatively stable when vehicles travel in the same direction, but when vehicles run in the opposite direction, the network topology will change very quickly, and the life of the whole link will be shortened. It is possible to predict the link state by combining the driving direction, speed and road information of the vehicle.

Nevertheless, VANET has the characteristics of openness, high-speed movement and dynamic topology change, which cause some transmission problems:

- The stability of wireless channels is poor, which is influenced by many factors, including the current road condition information, the relative speed and direction between vehicles, the types of vehicles, the buildings along the road, etc.;
- Rapid network topology change and short link life;
- Limited network capacity. The distribution of nodes in VANET is restricted by roads, and it presents a "tubular" shape. According to the calculation method of the network capacity of a random plane, it can be found that its network capacity is more limited than that of the general wireless mobile network;
- As the traffic density changes, the network load will also change greatly, so the nodes must have strong adaptability to this rapid change;
- VANET and driving safety are closely related, while OBU's operating environment is relatively harsh, which requires more stringent requirements for the reliability and safety of these devices.

## 2.2. Internet of Vehicle (IoV)

### 2.2.1. The Structure of IoV

In view of the defects of VANET, IoV has solved the problems existing in VANET. IoV refers to connecting vehicles to form a network, and it realizes the functions of intelligent transportation, intelligent vehicles and intelligent driving. In terms of networks, IoV is a three-tier architecture of "end management cloud", as shown in Figure 4 and Table 2.

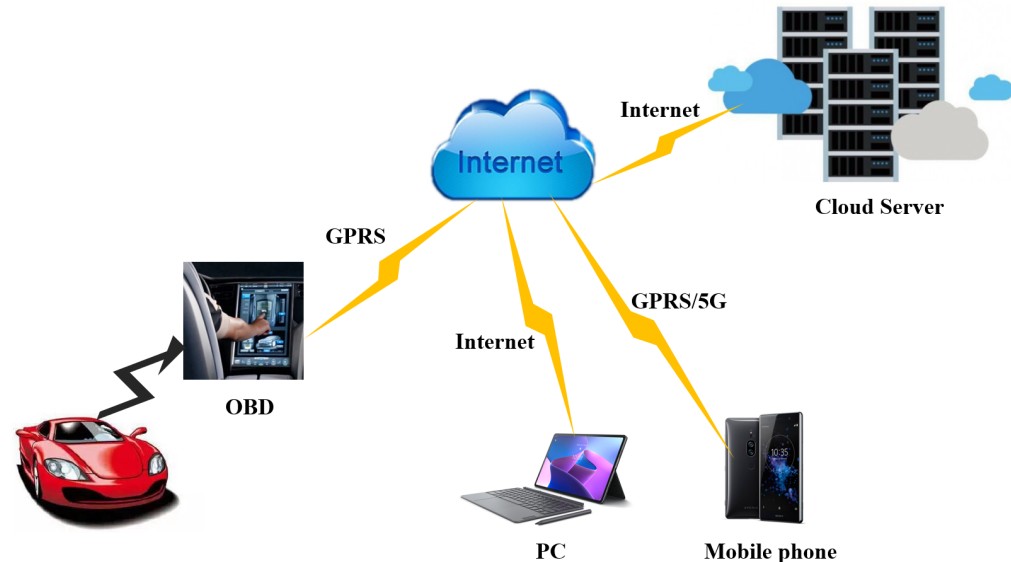

**Figure 4.** IoV system's "End Management Cloud" three-layer system.

**Table 2.** Architecture of IoV.

| Layer | Function |
| --- | --- |
| End | The end system is mainly used to collect and obtain relevant information about the vehicle, and it can sense the current driving state of the vehicle and the surrounding environment, which represents the intelligent sensing system of the vehicle. Moreover, it is also a ubiquitous communication terminal, just like V2X communication. Meanwhile, it is also a device that can identify the network credibly and enable the vehicle to have an addressing sum. |
| Management | This layer is used to solve the interconnection between V2X, realize the communication between vehicles and various heterogeneous networks, and ensure the serviceability, real-time performance and network universality from the aspects of function and performance. In the meantime, it unifies the private network and the public network. |
| Cloud | IoV is a cloud-based information platform, including ITS, logistics, mobile Internet, auto repair and auto parts, vehicle management, vehicle rental, insurance, vehicle management of enterprises and institutions, automobile manufacturers, emergency rescue, etc. The aggregation of multi-source mass information requires cloud computing functions such as mass storage, security authentication, virtualization and real-time interaction. |

### 2.2.2. Key Technologies of IoV

To realize the stable and efficient operation of IOV, many key technologies are needed:

- *Sensor technology and sensor information integration*—Sensor technology is mainly used to sense vehicle and road information, and it is mainly integrated in the OBU of vehicles. The sensor network of vehicles can be divided into in-vehicle sensor networks and out-of-vehicle sensor networks. The sensor network in vehicles is mainly used to provide information about the vehicle's status. This condition information is needed for remote diagnosis, for example, to analyze and judge the current state of the vehicle. Out-of-vehicle sensor networks are mainly used to sense the external environment of vehicles, such as cameras and anti-collision sensors. Such information can be used to assist driving, and it can also be used to improve driving safety. The road sensor network consists of sensors laid on roads and roadsides. Such sensors are used to sense road condition information and transmit it to vehicles, such as vehicle speed and direction, traffic density, intersection congestion, etc., so that the vehicle-mounted system can obtain road and traffic environment information.
- *An intelligent and open vehicle terminal system platform*—Based on a terminal system platform that is not sufficiently intelligent and open, it is difficult to build a network ecosystem. In this respect, we can see the importance of this point in the field of smartphones. At present, Google Android will become the mainstream operating system for IoV terminal systems. Being born for network applications, it is designed for touch operation, with rich applications, personalized customization, good user experience, and a huge increase in the number of applications, forming a mature network ecosystem.
- *Speech recognition technology*—This technology is very mature, which allows drivers to send commands to IoV through their mouths and receive services provided by IoV through their ears. This is most suitable for application in the fast-moving space of vehicles. The "cloud recognition" technology based on server-side technology must be used to solve the storage and computing capacity of vehicles, so as to adapt to the non-fixed command mode of speech recognition technology.
- *Server-side computing and service integration technology*—IoV uses cloud computing to plan the driving path of a large number of vehicles, analyze real-time road conditions, diagnose vehicles, and dispatch traffic congestion. IoV implements service innovation and provides value-added services through service integration.

- *Communication and its application technology*—IoV mainly depends on two communication technologies: short-distance wireless communication and long-distance mobile communication technologies. In the former, RFID sensors and WIFI-like communication technologies are predominant, while in the latter, GPRS, LTE, 5G and other mobile communication technologies are the main focus. These communication technologies are more concerned with applications, such as automatic toll collection, data packet transmission, video surveillance and so on.
- *Internet technology*—IoV can integrate the existing technologies and applications of the Internet and mobile Internet, but it is necessary to develop the characteristic Internet applications of IoV. Only in this way can more commercial benefits be brought to IoV.

On the whole, differences between VANET and IoV are shown in Table 1. IoV takes vehicles as nodes and information sources, and it connects the acquired information to the platform network through wireless communication and other technical means for analysis and management. Its core is information acquisition and feedback control, so as to realize the interconnection in V2X. With the increasing urban traffic congestion and the continuous progress of ITS technology, IoV has good development prospects. Just as PCs enter the Internet and mobile phones enter the mobile Internet, vehicles will surely enter the IoV and will go far.

## 3. Trust Management in the Internet of Vehicles

Due to the traditional security technology based on authentication and encryption mechanisms, we cannot effectively identify the internal attacks launched by authenticated nodes. Therefore, the trust management mechanism is becoming increasingly favored by scholars. Trust is an abstract concept. At present, there is no precise and widely accepted definition of trust. Different scholars have different understandings and definitions of trust. Gambetta [19] gave the following definition from the sociological point of view: "When we think someone is credible, it implies that we think that the probability that he will take actions beneficial to us in the future is large enough, so we can consider cooperating with him in some way. On the contrary, if we think he can't be trusted, it means that we think he has a low probability of taking beneficial actions, so we should avoid cooperating with him". Mayer [20] defined trust as follows: "the trustor is willing to bear the risks brought by trusting the trustor, regardless of whether it monitors or controls the behavior of the trustor, based on the expectation of a certain behavior of the trustor". Faragazzedin [21] has another definition of trust: "Trust is the belief in an entity, and within a certain period of time, this belief will change with the change of the entity's behavior". Alfzabdul-Rahman [22] defined trust as follows: "Trust is a reflection of the probability of an entity performing a certain action. Although the action can't be monitored and predicted, this reflection will affect the behavioral strategies adopted by people". Li [23] thought: "Trust is a subjective behavior between entities, and it is a summary judgment based on one's own experience and observed facts". Grandison [24] thought: "Trust is the belief of taking certain reliable behavior in a certain context".

### 3.1. Classification of Trust

In IoV, trust encompasses direct trust and indirect trust, in which direct trust and indirect trust are relative. As can be seen from Figure 5, when A and B are visible, B and C have direct trust, whereas when A and B are non-visible, B and C have indirect trust. Therefore, there is no absolute link between these two kinds of trust.

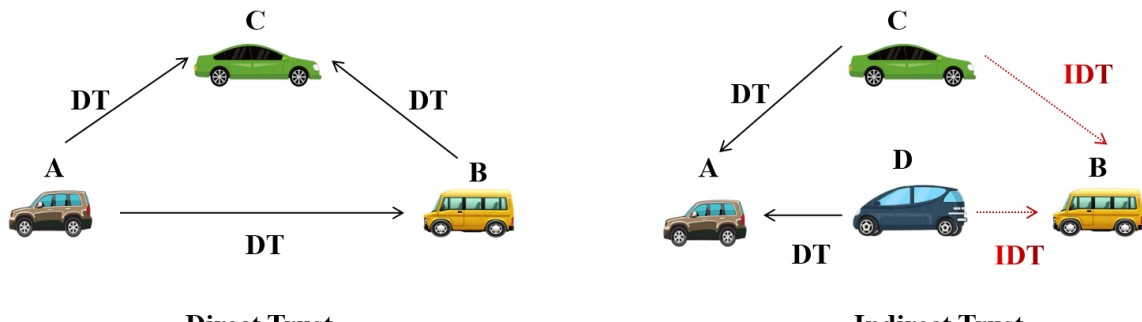

**Figure 5.** Classification of trust.

### 3.1.1. Direct Trust (DT)

Direct trust is the cognition formed by the accumulation of direct historical interactions between entities. The degree of trust between the trustor and the trustee is related to the degree of satisfaction with their historical interactions. Through the direct interaction between entities, credibility can be flexibly adjusted to reflect the dynamic change in trust. At present, there are many methods to calculate direct trust. Sarah [25] used the similarity, family and packet delivery ratio; in [26], the quality of received messages and the ability of nodes to disambiguate messages were used, as represented by Equation (1); and in [27], familiarity and package delivery rate were used to calculate direct trust, taking timeliness and interaction frequency as the weight of the trust calculation. Alnasser [28] calculated the trust value by using the forwarding rate of messages in time interval $t$, as shown in Equation (2); Ga [29] calculated direct trust by Bayesian inference and revised the trust value by penalty factor, as shown in Equation (3); Ji [30] used $H^t_{(j,k)}$ to represent the legal behavior of node $j$ to node $k$ in a specific time period $t$; and Mao [31] used inter-vehicle subordinate trust weight and the original trust of the vehicle.

$$DTR = \frac{1}{2}\sum_{i=1}^{n}\left(\frac{M_{Quality} \times MDR}{M_{Quality} + MDR}\right) \tag{1}$$

where $DTR$ is the direct trust, $M_{Quality}$ is the quality of the received message, and $MDR$ is the message dissemination ratio.

$$T^{(t)}_{d_{(i,j)}} = \frac{Successful - Interactions}{Total - Interactions} \tag{2}$$

where Successful–Interactions is the number of successful interactions between nodes, and Total–Interactions is the number of total interactions between nodes.

$$TR_{D_{ij}}(N+1) = \frac{N_s + 1}{N_s + N_f + 1}(1-\mu)^N f \tag{3}$$

where $N_s$ is successful forwarding times, $N_f$ is failed forwarding times, and $\mu$ indicates the sensitivity of forwarding failure.

### 3.1.2. Indirect Trust (IDT)

Indirect trust, also known as recommendation trust, is calculated from the recommendation information of other entities (such as neighbor nodes). Indirect trust indicates that trust has a certain transitivity. When the trustor evaluates the trust, they can not only make a comprehensive calculation based on the direct interaction records but also combine the recommendation information of other entities with the trusted party. Ahmad [26] used positive opinions and negative opinions, as shown in Equation (4). Alnasser [28] first calculated the confidence value between two adjacent nodes; then, they divided all

recommendations into positive recommendations and negative recommendations, and finally, they gave different weights to the two recommendations, as shown in Equation (5); Ga [29] used reputation to calculate the indirect trust value, as shown in Equation (6); Ji [30] used a cosine-based similarity metric and trust ratings and then used Resnick's standard prediction formula to calculate the recommended trust value; and Mao [31] used the role-based trust weight, the trust opinion of neighbors and the original trust of the vehicle;

$$ITR = \sqrt[n]{\left[\left(\frac{\alpha}{\alpha+\beta}\right) \times \sum_{i=1}^{n} P_o + \left(\frac{\beta}{\alpha+\beta}\right) \times \sum_{i=1}^{n} N_o\right]} \tag{4}$$

$$T_{in_{(i,j)}}^{(t)} = \alpha P_{(i,j)}^{(t)} + \beta N_{(i,j)}^{(t)} \tag{5}$$

where $P_{(i,j)}^{(t)}$ is the average value of the positive recommendations, and $N_{(i,j)}^{(t)}$ is the average value of the negative recommendations. $\alpha$ and $\beta$ are the weight of the two of them, respectively.

$$TR_{R_{ij}} = \frac{\Sigma TR_{Dk_j^r}}{R} \tag{6}$$

where $\alpha$ and $\beta$ are the reward and penalty factors, respectively. $P_o$ is positive opinions and $N_o$ is negative opinions.

*3.2. Trust Models*

3.2.1. Classification of Trust Models

IoV trust management mainly includes the establishment and evaluation of trust models. According to different evaluation objects, trust models can be divided into three categories, as shown in Figure 6: the entity-based trust model (ETM), the data-based trust model (DTM), and the composite trust model (CTM).

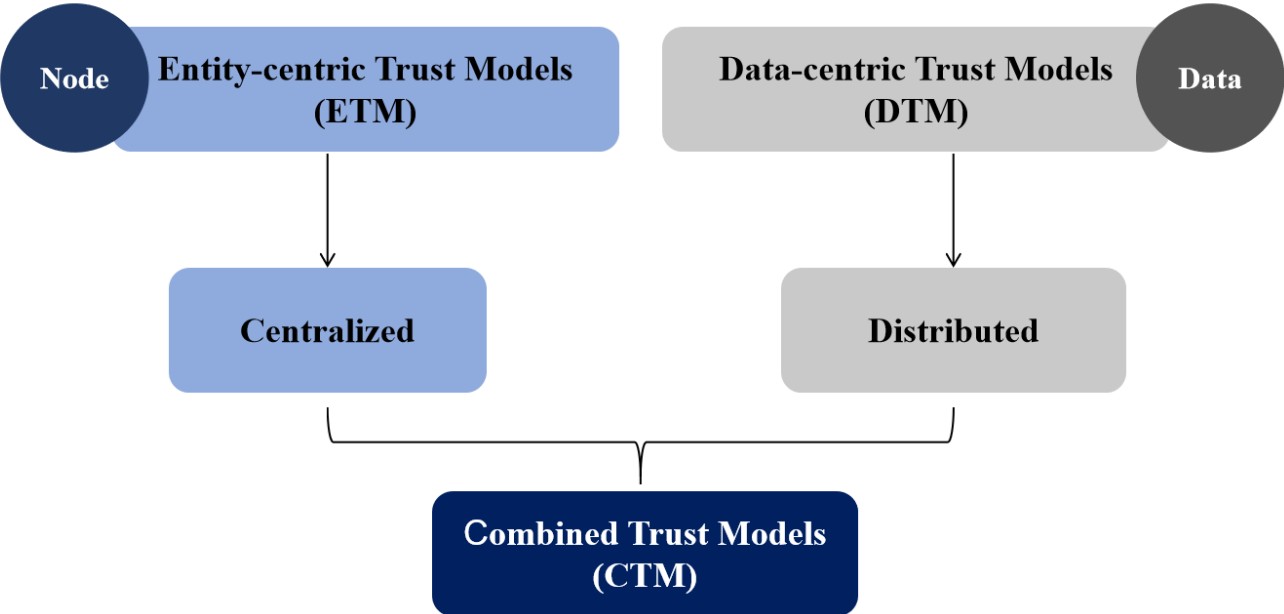

**Figure 6.** Classification of trust models.

- *Entity-based Trust Model (ETM)*
  The purpose of ETM is to evaluate the credibility of vehicle nodes. Usually, a trust evaluation system can be built by using direct or indirect trust between vehicles, and the trust value of each node can be calculated so as to detect untrustworthy vehicles. Hu [30] et al. put forward a trust model based on the feedback data of users' vehicles

and evaluated the credibility of the first vehicle in the vehicle arrangement scenario, so as to help the user's vehicle to choose a reliable vehicle to follow. In [12], the social attributes of nodes are used, and three social trust relationships, direct neighbor trust, indirect neighbor trust and friend trust, are considered. The trust degree of nodes is calculated by weighted average.

In [32], the main difficulty of an entity-based trust model maintaining the trust relationship between vehicles lies in how to collect enough information to evaluate the trust degree of nodes, especially when vehicles have just joined VANET. Additionally, when there is not enough interactive information, or there are few connections between vehicles and short communication links, e.g., in suburbs with sparse vehicles, these factors can make it difficult to evaluate trust effectively.

- *Data-based Trust Model (DTM)*

  DTM mainly evaluates whether the received data are reliable. The model needs to collect messages from various information sources, such as neighbor nodes and RSU, and filter out untrustworthy data to accurately verify the reliability of the received data. Tahani [11] established a distributed trust mechanism based on a direct experience survey between neighboring vehicles according to VANET's characteristics. Every vehicle first detects whether the received data are reliable, and then, it distributes the trust value to all of their neighbors. Rawat [24] proposed using the received signal strength (RSS) and the geographic location (GPS) of the vehicle to evaluate the trust level of the received message, combining the Bayesian estimation evaluation algorithm with the determined distance calculation method to provide better evaluation results, thus helping to identify malicious message data.

  The main disadvantage of DTM is that the trust relationship between vehicles can never be formed, and only a short-term trust of the received data can be established. Because data trust is based on events, it is necessary to build the trust relationship for each event again and again, and the previous trust data have not been used. At the same time, when the number of received messages is insufficient, it is difficult to judge the accuracy of the messages.

- *Combined Trust Model (CTM)*

  CTM is equivalent to the combination of the above two trust models. This model can not only evaluate the reliability of vehicles but also calculate the reliability of data. Generally, the two trust models are interrelated, that is, the trust value of a node affects the credibility of data to a certain extent, and the trust value of data in turn reflects the credibility of a node. An anti-attack trust management scheme (ART) was proposed by Li [11]: in this system, malicious attacks can be detected and responded, and the credibility of data and mobile nodes in VANET can be evaluated. Data trust evaluation is based on feelings and the data collected by multiple vehicles in two dimensions of node trust evaluation, namely function trust and recommendation trust. Mehmood [33] put forward a cluster-based hybrid VANETs trust management scheme, which can not only classify the information in the cluster but also identify malicious vehicles in real time.

In [34] CTM inherits the advantages and disadvantages of ETM and DTM. It not only needs to build a complete trust relationship between vehicles but also needs to be able to evaluate the credibility of each data point. Compared with the former two trust models, the trust evaluation process is more complicated, and the system overhead will be higher. At the same time, this model also has some limitations, such as sparse data.

The main function of the trust model is to transmit reliable, true, and effective information between V2X. There have been many articles about different trust models, as demonstrated in Table 3. A new context-based trust assessment and management framework was proposed in [19]. The authors tested five parameters of three trust models, EOTM, DOTM, and HTM, in four different contexts, and the four contexts were divided into mobility and attack types. This paper mainly considers the QoS and security of the system, tests some practical evaluation criteria, and uses the actual scenario of OpenStreetMap

to set up different scenarios in urban and rural areas. In [35], the authors put forward a task-based experience reputation (TER) framework, in which the fog node is used as the memory to store the updated trust value, and the authors also proposed two reputation-updating methods, which were compared by simulation experiments. Finally, this model was evaluated by using the overhead and workload of information transmission. In [36], the authors developed a quantifiable trust evaluation model based on machine learning. This paper focused on how to use the combination of trust parameters to find the optimal trust boundary so as to distinguish between trusted nodes and untrusted nodes. In [37], the authors put forward a risk-based trust evaluation advanced model (RTEAM). The model integrates risk into the trust model by calculating the possibility of performing incorrect behavior and the impact of such a possibility. This model was evaluated by UND and TPR, and other trust models were compared. In [38], the authors proposed a distributed trust model based on recommendation, which detected non-stable malicious behaviors. The model evaluation used two parameters, PDR and network throughput, which were compared with some existing trust models in the testing process, especially with regard to improving PDR. In [39], the authors put forward a new trust model, REK, which was used to simulate the process of human perception in the SIoT environment.

**Table 3.** A comparison of the existing state-of-the-art models.

| No. | Trust Parameters | Contributions |
| --- | --- | --- |
| [25] | Similarity (SMR), familiarity (FMR), packet delivery ratio (PDR) | Five algorithms in machine learning were used to analyze similarity, familiarity, and packet delivery ratio. |
| [28] | Confidence, interactions | This paper proposed a V2X communication trust model based on a recommendation to resist internal attacks. |
| [29] | Confidence, time sliding window, and time decay function | The author proposed a trust evaluation and management model based on the perspective of historical interaction. |
| [30] | Detection rate | Evidence combination-based collaborative trust management scheme against attacks. |
| [31] | Inter-vehicular subjective trust weight, role-based trust weight, original trust of vehicle, neighbor trust calculation | Multi-level Hybrid Trust Management Model (HHTM). |
| [40] | Resource availability, trust score | The author proposed a cluster-based hybrid VANETs trust management scheme. |
| [35] | Location, time closeness, information quality, confidence | Context-based trust assessment and management framework. |
| [36] | Reward | Task-based experience reputation (TER) framework. |
| [37] | Co-location relationship, co-work relationship, frequency and duration, cooperativeness, reward system, mutuality and centrality, community of interest | A quantifiable trust evaluation mechanism based on machine learning. |
| [38] | Likelihood, impact | Risk-based trust evaluation advanced model (RTEAM). |
| [41] | Reward, loss | A heterogeneous blockchain-based Hierarchical Trust Evaluation strategy. |
| [42] | Attitude (AT), subjective norms (SN), perceived behavioral control (PBC) | The author put forward a trust evaluation model based on human psychology. |
| [43] | Development, loss, decay | An integrated trust model, called REK, takes the third party's opinion, experience and direct observation as three trust indicators. |
| [44] | Friendship similarity, cooperativeness, co-work similarity, Community of Interest | The author proposed a time-aware trust model that utilizes social relationships. |
| [45] | Cooperation, freshness of data | Hybrid trust management mechanism based on communication and data. |

**Table 3.** *Cont.*

| No. | Trust Parameters | Contributions |
|---|---|---|
| [46] | Source's location, event location, rvent time | The author proposed a trust management scheme based on a crediting technique in MATLAB. |
| [47] | Information quality, role-oriented trust, effective distance | Hybrid trust management (NCT and DCT). |

### 3.2.2. Trust Parameters

Trust parameters are mainly used to calculate trust values. Many parameters are introduced in the process of trust calculation, and the representative parameters are shown in Table 1. The trust parameters with high utilization rates are similarity, familiarity, confidence, co-location relationship, packet delivery ratio (PDR), Community of Interest (CoI), and so on. Moreover, the same trust parameters can be calculated in different ways. For example, the calculation of Community of Interest (CoI) in [37,48] is as follows:

$$T_{C_{o_I}^{t(O_i,O_j)}} = \frac{|C_{o_i}||C_{o_j}|}{|C_{o_i}| \cup |C_{o_j}|} \tag{7}$$

where $C_{o_i}$ and $C_{o_j}$ depict the interest group of objects $i$ and $j$, respectively.

$$K_{i_j}^{CoI}(t) = \frac{|M_{i_j}^{CoI}}{M_i^{CoI}} \tag{8}$$

The authors defined $|M_{i_j}^{CoI}$ as the set of communities jointly owned by trustors and trustees, and $M_i^{CoI}$ as the set of communities with each including the trustee as a member. Moreover, confidence has different explanations in different papers, such as in [28,29]:

$$C_{(i,k)}^{(t)} = \begin{cases} 1 & \text{if } T_{d(i,k)}^{(t)} \geq ThC; \\ 0.8 & \text{if } ThT \leq T_{d(i,k)}^{(t)} \leq ThC; \\ 0 & \text{if } T^{(}t)_{d(i,k)} \leq ThT. \end{cases} \tag{9}$$

where $Th_C$ is the confidence threshold, and $Th_T$ is the trust threshold.

$$\gamma = \frac{\int_{TR_{Dij}-\epsilon}^{TR_{Dij}+\epsilon} p_s^{N-1}(1-p)p_f^{N-1}dp}{\int_0^1 p_s^{N-1}(1-p)p_f^{N-1}dp} \tag{10}$$

where $(TR_{Dij} - \epsilon, TR_{Dij} + \epsilon)$ is the confidence interval. The authors used Bayesian inference to calculate the confidence value.

- *Co-Location Relationship (CLR)*—The concept of CLR is that when the trustee and the trustor are very close to each other, the trustor can easily obtain the information needed from the selected trustee, because from the perspective of position, the trustee is more reliable than other objects that are far away.
  The decision boundary is set according to the distance from the trustor in order to avoid the vehicle leaving the predetermined physical location, as demonstrated in [37], which gathered many parameters such as CWR, RS, CoI, MC, etc., and used the machine learning algorithm to find an optimal boundary to distinguish between trusted and untrusted nodes.
- *Co-work Relationship (CWR)*—CWR describes the interaction between nodes that are service-dependent rather than physically adjacent. It is the association that one node possesses with another when providing a service, and it can be calculated

by multicast interaction. This association represents the relative amount of shared multicast messages in relation to the total messages sent [37].

- *Rewards*—This parameter is used to evaluate the historical rewards between the trustee and the trustor. The more reasonable the interactions, the higher the reward value. This parameter can track the misbehavior of the trustee and view their history, so that the trustor can determine whether to have further communication with the trustee [37]. In [41,49], using joint deep learning to evaluate the degree of trust between users and task assignors, the author designed a layered incentive mechanism to realize reasonable and fair rewards and punishments, which improved the accuracy of trust evaluation.

- *Community of Interest (CoI)*—CoI represents the social status of the trustee and describes whether there is a close relationship between the trustor and the trustee in the social network. This parameter indicates the degree of mutual interest between the trustee and trustor. In general, the higher the CoI between the two nodes, the better the interaction between them and the more trustworthy they are deemed to be [37]. In [48], the authors calculated the community-based trust characteristics of the trustee relative to the trustor at time t.

- *Mutuality and Centrality (MC)*—MC represents the position of the trustee relative to the trustor in the network. The larger the MC is, the more similar the social relationship between them, the more interactions between them, and the higher the degree of trust [37].

- *Confidence*—Confidence indicates the accuracy of the probability estimate. It can save network resources when the credibility of direct trust is greater than the set threshold, and a node is considered as a trusted node without calculating the recommended credibility [29].

- *Packet Delivery Ratio (PDR)*—PDR refers to the contact degree between the trustee and the principal, which is generally defined as the packet forwarding rate between nodes. PDR is generally considered a very important parameter in establishing a trust model, calculating the trust value and identifying malicious vehicles [26,50,51].

- *Similarity*—This parameter is used to measure the similarity in content and service between any two vehicles. Generally expressed as the Euclidean distance, the authors of [31] used the concept of similarity to calculate the similarity between the vehicle information in the infrastructure trust table and the information sent to the vehicle.

- *Familiarity*—Familiarity indicates the familiarity between the trustor and the trustee, which is a parameter with a high utilization rate. This parameter is used to measure the interaction frequency between the trustee and the trustor. The higher the interaction frequency, the more information can be obtained from the other party, and the more favorable it is to gain higher trust [25].

- *End-to-End Delay (E2ED)*—This parameter is related to the QoS of trust management, which describes the total delay caused by sharing data packets generated by legitimate vehicles with neighboring vehicles. Of course, the smaller the value of E2ED, the more reasonably the trust mechanism is designed [41].

### 3.2.3. Evaluation of Trust Models

There are two important steps in trust management: establishing a trust model and evaluating the trust model. The above discussion introduced the existing trust model in detail, and trust evaluation is also a crucial step. Up to now, most papers have used evaluation parameters to evaluate trust models; specific evaluation parameters are shown in Table 4. In general, trust evaluation parameters include: precision (P), recall (R), F-score (F), accuracy (A), TPR, TNR, FPR, FNR, etc., which are mainly used to evaluate whether the established trust model is feasible. For example, using the combination of multiple parameters to evaluate the model [52], this paper proposed a novel trust model: NOTRINO. In order to evaluate the trust model, the authors verified three parameters under several different attacks: precision, recall, and F-score. By comparison, the three parameters of

this model reached the highest values, which shows that this trust model was feasible and can be used to detect malicious attacks and ensure the safety of the IoV. In [47], the authors proposed a hierarchical hybrid trust management model (HHTM). In this model, three trust values need to be calculated and then mixed according to different weights. Three parameters are also used to evaluate the accuracy of the HHTM scheme: P, R, and F. The experimental results show that with the increase in the number of nodes, these three parameters all increased, and their values were higher than those of the basic model.

Every established trust model is evaluated in various ways. To date, there have been few evaluations of the network, with most of the existing papers failing to achieve this. Nevertheless, the network is very important to the trust management mechanism, so network evaluation is the next topic that we should focus on.

**Table 4.** Existing evaluation parameters and simulation methods.

| No. | Evaluation Parameters | Simulation Tools |
| --- | --- | --- |
| [11] | Precision(P), recall (R), communication overhead | GloMoSim 2.03. |
| [27] | FMR (familiarity), PDR | CRAWDAD dataset. MATLAB. |
| [28] | Recommendation usage rate, FNR, prediction rate | Undefined. |
| [29] | Influence of node behavior on direct trust value, influence of the integration of direct trust and recommended trust on PDR, influence of the time sliding window and time decay, function on the direct trust value | Mobile model. |
| [30] | Precision and recall | NS2. |
| [31] | Performance quality level (PQL), feedback accuracy level (FAL) | MATLAB. |
| [35] | End-to-end delay (E2ED), event detection probability (EDP), anomaly ratio (AR), FPR, trusted and untrusted packets | Veins. |
| [36] | FPR, TPR, precision, recall | Network Simulator 2.35 (NS 2.35), the Open Street Map (OSM) database. |
| [37] | FPR, TNR | SIGCOMM-2009 conference, which is available in CRAWDAD. |
| [38] | Undefined cases (UND), true positive rate (TPR) | MATLAB. |
| [53] | TPR, FPR, trust computation error | NS 2.35. |
| [54] | Accuracy, recall, precision, F1-measure | CICIDS2017 dataset MATLAB R2019a on Windows 10 (random forest and coresets models). |
| [45] | The rate of untrust packets, the rate of trust packets, packet delivery ratio | Network Simulator Omnet ++ and Veins (Vehicle in Network). |
| [46] | Travel time, accuracy, $CO_2$ emissions, communication overhead | Veins. |
| [55] | Pairwise orderedness, threshold of trust value, TPR, TNR | Veins. |
| [56] | Precision, recall, F1-score | CRAWDAD dataset. |

### 3.2.4. Trust Model Simulation Method

The evaluation of the trust model in VANET mostly adopts simulation methods. Specifically, most of these suggestions use the NS-2 simulator [42]. Additionally, some proposals adopt other existing simulation tools, such as GloMoSim 2.03 [11], Matlab [12,27,38,57,58],

and Veins [36,46,54,55,59,60]. In addition, some papers have used C++ or Java programming languages for simulation. Table 2 summarizes the simulation tools in detail. Of course, in some papers, the trust model is not evaluated by simulation but only by theoretical analysis and discussion. For example, in [41], the author proposed a global trust-building scheme based on reputation (RGTEs). This scheme introduces a solution to realize the secure sharing of trust information in VANET by using statistical rules, which makes it more efficient and accurate for VANET to build trust in a rapidly changing environment. In addition, according to the real-time reputation of the network, the author used a dynamic threshold to detect malicious nodes. Many previous studies only used hypothetical methods in the simulation stage, and the trust model was not evaluated in the real car networking environment, which are limitations that we should improve upon in the future.

*3.3. Categories of Attacks*

Due to the particularity of the IoV's environment and the lack of infrastructure, it is vulnerable to a large number of attacks, especially smart malicious attacks, which are often difficult to detect, because such malicious attacks switch back and forth between trusted and untrusted. Hence, we need to design a trust model with better accuracy and use various flexible and effective detection methods. Different trust models can detect different attack types, but the attack types in the system may change at any time, and the fixed trust model may not be detected; details are shown in Table 5. Designing a trust model that can detect any attack type is therefore also a challenge that we need to solve in the future. Trust-based attacks are classified as follows:

- *Bad-mouthing attack and Good-mouthing attack*
  Attackers attempt to send fake trust messages to frame legitimate nodes so that they are not detected. Hence, the purpose of this attack is to undermine the proper trust assessment and make malicious attacks hard to identify. In [41], the authors proposed a trust mechanism based on evidence combination, which can resist bad-mouthing attacks. The precision and recall were still higher than 80% under bad-mouthing attack. Compared with bad-mouthing attacks, good-mouthing attacks send positive recommendations about malicious nodes.
- *Selective Misbehavior Attack*
  During a selective misbehavior attack, malicious nodes only provide false messages to some nodes, which is normal for other nodes, which will lead to inconsistent trust among different nodes that is difficult to detect. During a time-dependent attack, the behavior of nodes changes with time and is not fixed. In [41], three types of attacks were detected: bad-mouthing attacks, selective misbehavior attacks, and time-dependent attacks.
- *Time-varying Attack*
  During a time-varying attack, the behavior of nodes changes with time and is not fixed. Initially, an attacker would establish itself as a legitimate node for a short period of time, gain the trust of other vehicles, and then launch an attack, sharing malicious messages and ratings with neighboring vehicles. In [47], the authors detected this kind of attack.
- *Zig-Zag Attack (On-and-off Attack)*
  During a zig-zag attack, also known as the "on-and-off" attack, malicious nodes attack randomly. At first, these nodes are normal, and when they gain enough trust values, they launch malicious attacks, which are difficult to detect. In [52], the authors proposed an effective attack detection model, which mainly detected man-in-the-middle (MITM) attacks and zig-zag attacks as well as a combination of the two attacks.
- *Self-promoting Attack*
  Unlike other types of malicious attacks, selfish attacks gain their own benefits from attacks, which indicates that there is little cooperation between vehicles. For example, in [55], the authors proposed an incentive technology to prevent such attacks.
- *Whitewashing Attack/Newcomer Attack*

During a whitewashing attack, a node has a bad history before entering the system, but after re-entering the system, the node adopts a new identity in order to gain more trust and erase its dark history. In order to resist such attacks, in [31], new nodes were given a relatively low trust value, and an adaptive attenuation factor was also introduced so that the trust value of newcomers could reach a relatively high value over a long period of time.

**Table 5.** Main existing attacks.

| No. | Attack Types |
| --- | --- |
| [11] | Simple attack (SA), bad-mouthing attack (BMA), zig-zag (on-and-off) attack (ZA). |
| [28] | Blackhole attackers, grayhole attackers. |
| [30] | Bad-mouthing attacks, time-dependent attacks, selective misbehavior attacks. |
| [41] | Task attack, privacy leakage attack. |
| [47] | Man-in-the-Middle (MiTM) attacks, zig-zag attacks. |
| [55] | On-and-off attack, newcomer attack, collusion attack. |
| [56] | Bad-mouthing attack, semi-honest attack. |
| [61] | Cheating attack, grayhole, bad-mouthing attack. |
| [62] | Self-promoting attacks, ballot-stuffing attacks, whitewashing attacks, bad-mouthing attacks, discriminatory attacks. |
| [63] | Certificate replication attack, eavesdropping attack, attacks on privacy. |

## 4. Open Research Directions

Whilst a considerable amount of research has been published over the past couple of years, illustrating the various facets of trust management vehicular networks, there are substantial challenges that demand immediate attention before the promising paradigm of trust can be embedded for strengthening the resilience of the IoV networks. Accordingly, some of these research challenges are delineated as follows:

### 4.1. Threshold Setting

The purpose of trust management is to divide the vehicles in the system into the categories of trustworthy and untrustworthy, so it is usually necessary to set a trust threshold. Those above the trust threshold are trusted nodes; otherwise, they are malicious nodes. Thus, to improve accuracy, it is extremely important to set a threshold. Either too high or too low a threshold will threaten the security of the system. However, the thresholds in the existing literature are all steady, but the vehicle network is a dynamic system, and it is necessary to set an adaptive threshold. The threshold value can be adjusted in real time with the number of vehicles, environment, and other factors.

### 4.2. Data Collection

IoV has a highly dynamic topology, and the calculation of trust is closely related to the historical interaction between vehicles. For example, in rural areas, there are fewer vehicles and less interaction between them, which is unfavorable for calculating the trust value. Therefore, it is necessary to establish a trust mechanism that can adapt to any vehicle environment, such as setting a dynamic data collection speed and collection range, reducing the collection speed and range for crowded cities, and increasing the collection speed and range for suburbs with few vehicles.

### 4.3. Standards Introduction

In [64,65], the authors raised concerns about standards of trust, but most articles do not mention such standards, especially in cloud computing and fog computing, e.g., in [66,67]. There is a need to establish trust standards to improve the security of trust models, especially evaluation standards. In particular, the detection of smart attacks according to evaluation standards, which can endanger the security of the whole system, is a basic key problem that needs to be solved.

### 4.4. Trust Computing Validation

Trust calculation is a very important part of trust assessment. First, the trust value is calculated, and then, it is compared with the threshold value to determine the malicious nodes. However, existing papers have not yet reviewed the calculated trust value nor how to cancel the calculated trust value when it is wrong, therefore making this another urgent problem to be solved.

### 4.5. Community Formation

In the future, the IoV could form a community of independent isolated vehicles through the IoT. Typically, an object in IoT collaborates with at least one community. The same can happen with the IoV. Similarly, if the trustee and the trustor are in the same community and have the same interest group, the extent and similarity of the trustee's and the trustor's common interest can be indicated, which is convenient for research and can save resources.

### 4.6. Integration of Blockchain and IoV

Blockchains can provide a large number of innovative solutions in most IoV application scenarios. In most IoV scenarios, many data are generated and exchanged. There are many classical technologies that cannot be effectively applied to IoV scenarios. In addition, in this scenario, increasing communication connections may cause security holes. Another aspect is that the introduction of blockchain in the IoV not only improves trust, security, and privacy but also improves system performance and automation. Therefore, in this scenario, the powerful technology of blockchain should be used to improve the flexibility of the system and the ability to process a large amount of data. However, as blockchain research is still in its infancy, not only is it complex to analyze, but there are still many issues to be solved, relating to regulatory and legal concerns, performance, limited storage, security and privacy, optimized consensus, and incentive mechanisms [68].

## 5. Conclusions

The Internet of Vehicles is an important component of the smart cities and intelligent transportation that countries around the world are presently striving to develop. The IoV is a system integrating people, vehicles, things, and ambients, which links all kinds of information from inside the vehicles, outside the vehicles, and the environment to form a broad and intelligent network system. This paper focused on the reasons for the transition from VANET to IoV, the existing trust management models and parameters, and the analysis of the key problems to be solved in the future of IoV, laying a foundation for future research. To sum up, this paper provided some guidance for the future study of trust management mechanisms in IoV.

**Author Contributions:** Conceptualization, Y.W., H.Z., M.F.M.S., X.W. and L.C.K; investigation, Y.W. and X.W.; methodology, Y.W. and H.Z.; validation, Y.W. and H.Z.; writing—original draft preparation, Y.W. and H.Z.; writing—review and editing, Y.W., H.Z. and X.W.; supervision, H.Z., M.F.M.S., X.W. and L.C.K. All authors have read and agreed to the published version of the manuscript.

**Funding:** This research received no external funding.

**Informed Consent Statement:** Informed consent was obtained from all subjects involved in the study.

**Data Availability Statement:** Not applicable.

**Acknowledgments:** Wang Yingxun's research work is funded by the Universiti Malaysia Sarawak, Malaysia (UNIMAS) and the Qilu Institute of Technology, Shandong, China. The authors, therefore, acknowledge the UNIMAS and the Qilu Institute of Technology for generously sponsoring the research at hand.

**Conflicts of Interest:** The authors declare no conflict of interest.

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
