# Peer review of "Towards Strengthening the Resilience of IoV Networks—A Trust Management Perspective"

_futureinternet, doi:10.3390/fi14070202_

Round 1

Reviewer 1 Report

English Language must be improved.

The paper reads like a review of available models, proposals, and definitions. Although a lot of related research is cited, it is unclear how the different parts are related to each other, what advantages or problems emerge, and what the overall purpose of the review is. It looks like the purpose is to list the future directions for research, so the paper reads more like a research proposal rather than a piece of concrete research.  Perhaps it would be more appropriate as a feature article rather than as a research paper?

Some point specific comments.

1.       Keywords: Open research directions is not an appropriate keyword. Perhaps replace it with Smart Cities?

2.       Acronym V2X line 37 and SDN line 38 not explained.

3.       The description of related work lines 37-44 is a bit cursory and the reader cannot appreciate what the point is.

4.       What is e 3GPPTR?

5.       Include the Blaze 1996 reference mentioned in the paper.

6.       Figures 1, 2, 3, 4 should change location to be closer to the text where they are explained. Also table 1 is far down from where it is explained. Generally revise the location of all figures and tables.

7.       Not all symbols that appear in equations (1)-(3) are explained.

8.       The future research directions section makes some claims that are not supported with references to the relevant literature. E.g. “There are some existing articles about trust standards, but most of the articles don’t  involve them, especially in cloud computing and fog computing”. What are these existing articles? In point 6 that relates to Blockchain integration with IoV there are lots of proposals in the literature. Are these lacking somewhere?

Reviewer 2 Report

In this manuscript, the authors propose a trust management perspective of IoV networks. In particular, the authors offer an overall study of methods and properties of trustworthiness management models along with evaluation and simulation approaches. Also, the authors shed the light on some open research directions in the context.

The proposed manuscript offers a detailed overview of IoV trust management models. The manuscript is well written and it flows well. It covers several aspects and it is represents a good resource for the topic. However, there are major presentation issues which, in the current state, cannot be overlooked. In particular, the authors could consider to address these issues in order to improve the quality of the manuscript. These are given below:

- A lot of studies have been produced in the context of trust definition and management in IoT. The authors successfully cover related works and provide a satisfactory overview of the related studies. However, this overview could be organized better. As an example, a table could be used to organize the background, the concepts and the aspects reported (similar to Tables 1 and 2).
- A work providing trust management via the definition of a property called "scope" has been introduced in [https://doi.org/10.1016/j.pmcj.2020.101223]. The authors could consider to cite it in the manuscript.